# Eosinophilia and Kidney Disease: More than Just an Incidental Finding?

**DOI:** 10.3390/jcm7120529

**Published:** 2018-12-08

**Authors:** Philipp Gauckler, Jae Il Shin, Gert Mayer, Andreas Kronbichler

**Affiliations:** 1Department of Internal Medicine IV (Nephrology and Hypertension), Medical University Innsbruck, 6020 Innsbruck, Austria; gert.mayer@i-med.ac.at (G.M.); andreas.kronbichler@i-med.ac.at (A.K.); 2Department of Pediatrics, Yonsei University College of Medicine, Seoul 03722, Korea; shinji@yuhs.ac; 3Department of Pediatric Nephrology, Severance Children’s Hospital, Seoul 03722, Korea; 4Institute of Kidney Disease Research, Yonsei University College of Medicine, Seoul 03722, Korea; 5Division of Cardiology, Emory University School of Medicine, Atlanta, GA 30322, USA

**Keywords:** eosinophilia, CKD, kidney disease, AKI, autoimmune disease, EGPA, interstitial nephritis, IgG4-related disease, vasculitis

## Abstract

Peripheral blood eosinophilia (PBE), defined as 500 eosinophils or above per microliter (µL) blood, is a condition that is not uncommon but often neglected in the management of patients with chronic kidney disease (CKD), acute kidney injury (AKI), or patients on renal replacement therapy (RRT). The nature of PBE in the context of kidney diseases is predominantly secondary or reactive and has to be distinguished from primary eosinophilic disorders. Nonetheless, the finding of persistent PBE can be a useful clue for the differential diagnosis of underdiagnosed entities and overlapping syndromes, such as eosinophilic granulomatosis with polyangiitis (EGPA), IgG4-related disease (IgG4-RD), acute interstitial nephritis (AIN), or the hypereosinophilic syndrome (HES). For patients on RRT, PBE may be an indicator for bio-incompatibility of the dialysis material, acute allograft rejection, or *Strongyloides* hyperinfection. In a subset of patients with EGPA, eosinophils might even be the driving force in disease pathogenesis. This improved understanding is already being used to facilitate novel therapeutic options. Mepolizumab has been licensed for the management of EGPA and is applied with the aim to abrogate the underlying immunologic process by blocking interleukin-5. The current article provides an overview of different renal pathologies that are associated with PBE. Further scientific effort is required to understand the exact role and function of eosinophils in these disorders which may pave the way to improved interdisciplinary management of such patients.

## 1. Introduction

The evidence of peripheral blood eosinophilia (PBE), defined as 500 eosinophils or above per microliter (µL) blood, is a condition occasionally seen and often disregarded in the management of patients with chronic kidney disease (CKD), acute kidney injury (AKI), or patients on renal replacement therapy (RRT), either in the context of renal disease or as a consequence of co-morbidities. Diagnostic efforts to determine the etiology of moderately or markedly elevated eosinophil counts are rarely made. This might be explained by the lack of therapeutic consequences, lack of knowledge of eosinophil function and the uncertain diagnostic value of PBE. Classical diseases associated with eosinophilia are atopic conditions such as allergic asthma, autoimmune processes, parasitic infections, and neoplastic disorders. 

Although many questions remain regarding the biology of eosinophils, a clearance function in parasitic infections has been postulated [1]. In addition, eosinophils may expand and play a role in a broad range of local and systemic inflammatory diseases [1]. However, only little is known about the functional and prognostic impact of eosinophils in the etiology and progression of these disorders. In the past few years, considerable progress was made in the molecular understanding of different, ‘immunologic-driven’, systemic disorders that are associated with both, kidney damage and eosinophilia. In fact, various systemic disorders affecting the kidney function may be accompanied by eosinophilia. The discovery of molecular mechanisms that are responsible for specific disease processes has led to a better understanding of eosinophil biology and paved the way for better diagnostics and the development of novel, more specific drug therapies.

In this article, we provide an overview of different renal pathologies that are associated with PBE. A particular focus is set on the meaning of PBE for the diagnosis of underdiagnosed entities, in order to improve the interdisciplinary management of such patients.

## 2. Eosinophilia, Hypereosinophilia and the Hypereosinophilic Syndrome (HES)

For an in-depth understanding of terminology, it seems to be important to emphasize that definitions and classifications in the field of eosinophilic disorders have changed over the years. Recent recommendations are based primarily on proposals delivered by the World Health Organization (WHO) and the International Cooperative Working Group on Eosinophil Disorders (ICOG-EO) [2,3]. The focus of our work is set on renal pathologies that are associated with PBE. Several classical eosinophilic disorders, including primary hematologic disorders such as eosinophil leukemia, may sometimes also affect the kidney but do not regularly induce persistent kidney damage. For the sake of simplicity, an overview of these eosinophilic disorders is given in Figure 1 and the current definition of HE and HES, as proposed by the ICOG-EO, is highlighted in Table 1. 

### 2.1. Biology of Eosinophil Granulocytes

Eosinophils can enter extravascular tissues and participate in various immunologic reactions relevant to innate immunity. Eosinophil precursors differentiate in the bone marrow in response to different cytokines, such as interleukin (IL)-3, granulocyte-macrophage colony stimulating factor (GM-CSF) and, most importantly, IL-5 before they transit through the blood into various organs, including the thymus, spleen, lymph nodes, uterus, and the gastrointestinal tract. Eosinophils produce and store different bioactive molecules, like eosinophil peroxidase (EP), eosinophil cationic proteins (ECP), and major basic protein (MBP) in their granules. Various conditions facilitate a type 2 helper T (Th2) cell and cytokine-induced activation of eosinophils leading to a release of these molecules with consecutive effects in the affected tissues [4,5]. MBP exerts cytotoxicity to various tissue-adjacent cells. ECP are ribonucleases with antiviral activity and EP is involved in the production of reactive oxygen species (ROS) to digest extracellular pathogens [6]. Other released bioactive molecules like leukotriene C4, platelet activating factor and prostaglandins mediate effects on the tonus of smooth muscle cells, vascular permeability, platelet aggregation and chemotaxis resulting in enhancement, regulation, and repair of inflammation [6,7]. Eosinophils also produce and secrete DNA traps which may be involved in the defense against various pathogens [1]. For a detailed review of the biology and physiology of eosinophils we refer to recent review articles [1,2,4,5,6].

### 2.2. Definitions and Classification of Eosinophil Disorders

Conditions with an eosinophil count (EC) >500/µL blood are generally referred to as PBE, whereas marked and persistent (examined in two occasions with a minimum of four weeks interval) elevations >1500/µL are categorized as hypereosinophilia (HE). Tissue eosinophilia meeting criteria depicted in Table 1 is also regarded a feature of HE [2]. A percentage greater than 5% of all leukocytes in peripheral blood is commonly regarded as pathological, but relative eosinophil counts are usually not used for diagnostic purposes because the number varies with the total white blood count (WBC). An arbitrary sub-classification, regarded as mild (>350–500/µL), modest (>1500/µL), and severe (>5000/µL) eosinophilia has been proposed [3,7,8,9]. 

Different circumstances can lead to an increased proliferation of eosinophils (Figure 1): primary hematologic disorders due to proliferation of neoplastic eosinophils and their progenitor cells have to be distinguished from secondary or reactive eosinophil expansion, induced by a Th2-mediated immunologic cascade of different cytokines (mainly IL-5, but sometimes also IL-3 and GM-CSF) resulting in an accumulation of polyclonal eosinophils [10]. After exclusion of primary and secondary causes, the condition is termed idiopathic (hyper-) eosinophilia. If, in such case, the criteria of HE are fulfilled but no clinical symptoms or organ dysfunction are seen, the term HE of undetermined significance (HE_us_) is appropriate [2]. 

In 1975, Chusid et al. introduced the term hypereosinophilic syndrome (HES) for cases with end-organ manifestations that are attributable to HE [11]. Chusid’s criteria were used for the definition of HES for nearly four decades and build the groundwork for the current definition, proposed by the WHO and ICOG-EO (see Table 1) [2,3]. Whereas there is consensus about the definition (criteria) of HES, a few open questions remain regarding the classification of rare eosinophilic disorders [9,12]. 

## 3. Eosinophilia in Kidney Disease

### 3.1. Kidney Involvement in the Hypereosinophilic Syndrome

There is a paucity of data linking idiopathic HES to conditions involving the kidney [13]. No case of renal involvement was reported in 50 patients with HES_I_ [14]. A classical complication of HES is Loeffler’s endocarditis which is characterized by progressive heart failure with endomyocardial fibrosis due to eosinophilic infiltration and secondary organ damage provoked by thromboembolic occlusion of larger and smaller vessels [15]. In general, all organ systems can be affected by HES. In a more recent retrospective analysis of HES cases, including primary and secondary (hematologic) forms of HES, skin involvement (69%) was most common, followed by pulmonary (44%), gastrointestinal (38%) and heart (20%) manifestations [16]. Former articles on idiopathic HES describe renal involvement in 0% [14] to 36% [11]. Renal affection seems to occur mostly late in the course of HES, predominantly due to vascular events as a result of thromboembolism or atheroembolism [11,17]. In this regard, it is worth noting that thromboembolism is regarded a major organ manifestation (clinical feature) of HES. A review of cases with HES_I_ and renal involvement identified certain primary renal parenchymal diseases in association with HES_I_. Eosinophilic interstitial nephritis, described as the most common pathologic pattern, followed by infrequent reports of membranous nephropathy (MN) and different other forms of glomerulonephritis, thrombotic microangiopathies (TMA), electrolyte disturbances and the presence of Charcot–Leyden crystals, are further kidney-defined disorders associated with HES_I_ [18].

### 3.2. Kidney Diseases Associated with Eosinophilia/Hypereosinophilia

An overview of chronic or acute kidney diseases and special conditions that are associated with eosinophilia is given in Table 2 and a review of these pathologies is provided in the following sections.

#### 3.2.1. Hypersensitivity Reactions

Eosinophilia is a common feature of hypersensitivity reactions, driven by antigen-activated Th2 cells and a consecutive release of critical cytokines. In all these reactions, IL-5 is considered the most relevant trigger of eosinophil proliferation, accumulation, and function. Basically, any exogenous antigen can provoke a hypersensitivity reaction with possible affection of the kidney function. 

Tubulointerstitial nephritis (TIN) is the histologic pattern found in acute interstitial nephritis (AIN) and other immune-mediated kidney diseases. AIN is found in about 5–18% of kidney biopsies performed in the setting of AKI and an increasing tendency in prevalence is seen in the last years [46]. Drug-induced AIN accounts for about three-quarters of the cases and basically any drug can trigger the disease. Antibiotics, non-steroidal anti-inflammatory drugs (NSAIDs) and proton-pump-inhibitors (PPIs) are the most frequent agents leading to AIN. Other possible causes of AIN are infections and autoimmune diseases like sarcoidosis, Sjögren syndrome, IgG4-related disease (IgG4-RD), or the tubuloinsterstitial nephritis and uveitis (TINU)-syndrome. Blood eosinophilia is seen in about 20% of the cases [19]. Urinalysis may reveal unspecific results like mid-range proteinuria, hematuria, pyuria, or abnormalities in the tubular function including Fanconi syndrome or renal tubular acidosis (RTA). Urinary eosinophils appear to be a poor diagnostic predictor with a rather low positive-predictive value, irrespective of using the traditional cut-off of >1% or a higher cut-off of >5% [19,47]. The classical clinical hypersensitivity-triad of fever, skin rash, and eosinophilia is rarely found and considering the subtle clinical signs and the multitude of possible triggers, it is not surprising that only in about half of the cases AIN was the suspected diagnosis before biopsy [19,48]. Potential triggers, especially drugs, should be withdrawn if possible. In the case of uncertainty or in absence of clinical improvement after drug withdrawal, an early kidney biopsy should be performed to verify the diagnosis. The role of corticosteroids in the therapy of AIN is controversially discussed [19,49,50,51]. Until now, no clear recommendation can be made, but two recent large retrospective studies suggest a benefit of steroid treatment. Prendecki et al. showed an improvement of GFR and a lower prevalence of ESRD after 6 and 25 months [52]. Fernandez-Juarez et al. showed a greater recovery of kidney function at six months when steroid treatment is started early [53]. General outcome and prognosis of AIN depends on the delay until the causative trigger is identified and removed as well as the degree of tubular atrophy and interstitial fibrosis on biopsy. Approximately half of the patients show full recovery, while the other half develops CKD or, in a small number, remains on RRT [19,20]. Relapses may occur after terminating an initially successful corticosteroid therapy. In some of these patients, an underlying systemic disease may be diagnosed. Drug-induced AIN can evolve into corticosteroid-dependent AIN even when the causative agent has been stopped [46].

A specific condition is the drug reaction with eosinophilia and systemic symptoms (DRESS) syndrome, which is a potentially life-threatening hypersensitivity reaction to a medication or its reactive metabolites. The precise pathogenesis is not fully understood and different theories like an underlying versus a secondarily-induced reactivation of different herpes viruses was proposed [54,55]. Among the known causative drugs, aromatic anticonvulsants (e.g., carbamazepine and phenytoine), allopurinol, sulfasalazine, trimethoprim-sulfamethoxazole, PPI, vancomycin, and minocycline are most frequently reported [56,57]. Clinical features, beside the typical cutaneous eruptions, are systemic symptoms including fever, eosinophilia, and different organ manifestations. PBE is present in the majority of the cases, whereas marked eosinophilia is described in approximately 30% [23,58,59]. The prevalence of renal involvement is ranging between 10% and 57% with highest rates described in association with allopurinol [21,22,23]. The histological pattern and the clinical findings are similar to those seen in AIN. Identification and withdrawal of the causative drug is the principal therapeutic challenge in DRESS. Systemic corticosteroids (suggested starting dose 1 mg/kg/day of prednisone or equivalent with slow tapering over 6–8 weeks) and a supportive therapy to control hemodynamics, fever and cutaneous symptoms are indicated, while prophylactic antibiotic therapy should be avoided [59]. Some patients show no response or have contraindications towards corticosteroid treatment. In this setting, alternative immunosuppressive or immunomodulatory therapeutic measures such as intravenous immunoglobulins (IVIG), plasma exchange, cyclophosphamide, cyclosporine, mycophenolate mofetil, and rituximab (RTX) have shown efficacy in selected cases [60,61,62].

Among 824 patients receiving outpatient parenteral antibiotics, 25% developed eosinophilia in a prospective cohort study. Most patients did not develop hypersensitivity reactions, while DRESS occurred in 0.8%. Eosinophilia increased the risk of developing renal injury (HR = 2.13, *p* = 0.0009) [63]. Drug-induced kidney injury seems to be an underestimated entity and eosinophilia might be a helpful diagnostic tool in these patients.

#### 3.2.2. Autoimmune Reactions and Related Diseases

The main vasculitic glomerulopathy associated with eosinophilia is eosinophilic granulomatosis with polyangiitis (EGPA, Churg-Strauss syndrome). EGPA differs from other anti-neutrophil cytoplasmic antibody (ANCA)-associated vasculitides (AAV), since patients have severe asthma as well as blood and tissue eosinophilia [26]. The American College of Rheumatology diagnostic criteria are asthma, eosinophilia, mono-polyneuropathy, pulmonary infiltrates, non-fixed paranasal sinus abnormalities, and extravascular eosinophils with at least four of six criteria to be fulfilled for diagnosis [64]. Patients with EGPA show a low frequency of ANCA-positivity (30–40%) with myeloperoxidase (MPO)-ANCA predominance [24,65,66,67]. ANCA-positivity is associated with a higher prevalence of renal disease [24,25], involvement of the upper airways and peripheral neuropathy. ANCA-negative EGPA, in contrast, shows more cardiac manifestations (including pericarditis and cardiomyopathy) and pleural effusions which are both usually associated with eosinophil-mediated disorders [24,25,65]. The linkage between ANCA-positivity and glomerulonephritis has been implemented in the revised Chapel Hill consensus conference nomenclature of vasculitides in 2012 [68]. More recently, two different disease phenotypes of EGPA have been discussed and different underlying pathophysiological processes suggested, but until now there is not enough evidence to split EGPA into different disease entities (see Figure 2). In fact, EGPA should be regarded as one syndrome with different phenotypes [69]. AAV have a clear genetic background and two genome-wide association studies (GWASs) showed distinct human leukocyte antigen (HLA) gene-variants for the three AAV-subtypes [70]. Vaglio et al. found the HLA-DRB4 gene and HLA-DRB1*07 allele as genetic risk factors for EGPA, correlating with a vasculitic phenotype and ANCA-positivity [71]. Moreover, IL-10.2 haplotype is associated with ANCA-negative EGPA and an increased IL-10 production [72]. A GWAS in the specific context of EGPA is ongoing and promises new insights into the pathophysiological understanding of this Janus-like disease.

Nasal involvement in GPA is characterized by epistaxis, crusting, rhinorrhoe, and classical signs of sinusitis [83], while extensive nasal polyposis or rhinitis (either allergic or non-allergic) are the hallmarks of nasal involvement in EGPA [84]. However, polyangiitis overlap syndrome (POS) of GPA and EGPA was suggested in a few cases that fulfilled diagnostic criteria (e.g., ACR criteria) for both, GPA and EGPA [85,86,87,88]. In respect thereof, patients with typical vasculitis lesions, marked eosinophilia and PR3-ANCA positivity may be classified as POS in the absence of asthma. The optimal therapy for POS is not known but it seems important to identify these patients. Long term high dose steroid exposure to control disease activity is necessary in many patients with EGPA [77], which is different to patients with GPA. 

On the other hand, ANCA-negative EGPA might be difficult to separate from HES_I_ because of overlapping clinical presentation and biomarker profiles [89]. Findings like the recent publication of a FIP1L1-PDGFRA-positive case of EGPA support the idea of overlapping syndromes that should be characterized by clinical appearance together with their biomarker profiles rather than finding an explicit diagnosis-label [90]. Given the fact that EGPA is a heterogeneous entity with potentially overlapping diseases carrying different biomarker profiles, patients could benefit from different treatment options like RTX for ANCA-positive EGPA, omalizumab or mepolizumab for ANCA-negative EGPA and imatinib for EGPA patients with mutations in TK fusion genes [91]. Testing for FIP1L1-PDGFRA mutation should therefore be performed routinely in all cases with HE, regardless of clinical presentation, suspected EGPA or ANCA-status. [92]. 

In systemic lupus erythematosus (SLE), kidney involvement in form of lupus nephritis occurs in about half of the patients [27]. PBE is a rare finding in subjects with SLE. Moreover, differential blood count has not been measured in most studies [28]. Harvey et al. reported PBE of 3% or more in 15 of 46 patients with SLE but only two patients had absolute eosinophil counts over 400/µL [93]. Reports of marked eosinophilia in association with SLE are restricted to single-case reports of SLE with concomitant Loeffler endocarditis [94,95], Wells’ syndrome [96], eosinophilic pleuritis [97], or EGPA and HES [98]. 

Kargili et al. explored the prevalence of eosinophilia in other rheumatologic diseases in a retrospective analysis of 1000 cases between 2001 and 2002. Eosinophilia was seen in 26 of 293 patients with fibromyalgia (8.9%), 3 of 182 patients with rheumatoid arthritis (1.65%), and none of 12 patients with scleroderma, but the low prevalence might be explained by the frequent use of corticosteroids among these patients [99]. A recent study analyzed the eosinophil count to lymphocyte ratio (proportion of eosinophils to lymphocytes within the leukocyte count) in 344 patients with SLE and observed lower eosinophil counts in subjects with SLE compared to other autoimmune disorders. In line, the eosinophil-lymphocyte ratio in cases with SLE was lower compared to the diseased and healthy controls [100].

#### 3.2.3. Vascular Diseases 

Thrombotic microangiopathy (TMA) is characterized by a clinical presentation with microangiopathic hemolytic anemia, thrombocytopenia, and frequent renal and neurological involvement. TMA may occur in many disease processes but is typically associated with hemolytic uremic syndrome (HUS) and thrombotic thrombocytopenic purpura (TTP) [101]. Renal involvement is typical in HUS and rather uncommon in TTP. The overall prevalence and the outcome of AKI in TMA is therefore determined by the TMA subtype [29]. Low evidence exists for an association of PBE and thrombotic microangiopathies (TMA). Eight reported cases of TMA with PBE were diagnosed as HES [17,102,103,104,105,106]. Recently, one case of atypical HUS with AIN and PBE was reported and the first two case reports of concomitant atypical HUS and EGPA were published [107,108]. Activated eosinophils are known to favor thrombosis by various mechanisms. Endothelial cell damage is supposed to be mediated by degranulation of bioactive molecules, like EP, ECP, and MBP. Additionally, these molecules showed direct effects on coagulation, e.g., by inactivation of endothelial cell-expressed thrombomodulin and induction of tissue factor activity in endothelial cells [109,110]. Nonetheless, this observation appears interesting because thrombosis is a major link between HE and eosinophil-mediated organ damage observed in HES [109]. Whereas arterial thrombosis shows a higher incidence in HES, EGPA appears to have a predominance of venous thrombosis [109]. Vascular thrombosis is the main clinical feature of HES and also seen in EGPA, whilst TMA is less frequently reported in these patients. Nonetheless, the co-existence of HE with TMA or with any other thrombotic event should be sufficient to diagnose HES.

A primary non-renal disease that is worth noting because of its frequent impact on kidney function and association with eosinophilia is the cholesterol embolization syndrome (CES). It is a systemic disease with distal showering of cholesterol crystals out of atheromatous plaques from the aorta or other major arteries. CES has a male preponderance, present at an older age (>50% over 70 years), and shares the same risk factors with atherosclerosis. CES can occur spontaneously but is mainly induced iatrogenic, e.g., after aortic surgery or arterial invasive procedures such as angiography or left heart catheterization. Moreover, the use of anticoagulation (both, heparin and oral anticoagulants) have been linked with CES [111,112]. Beside skin lesions (commonly livedo reticularis limited to the lower limbs and trunk), the kidney is the most common site of involvement, followed by the spleen, pancreas, gastrointestinal tract, and adrenal glands [111,113]. Atheroembolic renal disease (AERD) is seen either in the form of AKI by acute rupture of unstable plaques or in a chronic but progressive way due to slow release of eroded atherosclerotic lesions [112]. In a prospective study, Fukumoto et al. found CES in 1.4% of 1786 patients who underwent left-heart catheterization. 64% of the reported CES-cases had renal damage. Blood eosinophil count was significantly higher than in patients without CES but did not reach the threshold for HE. Interestingly, CES with renal involvement showed a greater increase of eosinophil counts (from 220/µL to 535/µL) compared to CES without renal dysfunction (from 302/µL to 339/µL) [114]. In general, the presence of eosinophilia in CES is ranging from 14% to 71% [31,32]. The overall incidence of renal involvement in CES was 92.2% in a review of 22 studies [30]. CES is possibly an underdiagnosed entity because biopsy is rarely performed and clinical signs of CES are often subtle, especially when the disease progress is slow. An autopsy study showed a higher prevalence (25–30%) of CES after angiography [115]. Recent estimations by Mayo and Swartz revealed that 5–10% of all AKIs could be due to atheroembolism [116]. Among cardiac patients, the stage of CKD seems to correlate positively with the peripheral eosinophil count, so subclinical cholesterol embolization might play a role in this subset of patients [117]. Light microscopy of renal biopsy shows atheroemboli in interlobular and arcuate arteries, but also in arterioles and glomeruli with typical appearance as lance-shaped clefts, due to dissolution of cholesterol crystals during formalin fixation [113]. Given the high frequency of skin involvement, biopsy of skin lesions is suggested for diagnostic purposes [111]. The overall prognosis is poor with mortality rates ranging from 58% to 90% and RRT is necessary in about half of the patients [30,112]. A specific therapy for CES is not established. Withdrawal of anticoagulants, if possible, is proposed by some authors, while the role of steroids remains controversial [111,118]. Lipid-lowering therapies with statins or low-density lipoprotein (LDL)-apheresis are being used in case reports but until now there are no randomized trials that support this strategy [119,120]. Considering the lack of therapeutic options for CES, prevention of further episodes is crucial. This includes the limitation of anticoagulation and the restriction of vascular-invasive procedures for patients at risk as far as possible [111]. 

#### 3.2.4. Other Diseases Presenting with Eosinophilia

Kimura’s disease (KD) is a rare benign chronic inflammatory disorder mainly affecting young Asian men and commonly presenting with recurrent subcutaneous masses (of head and neck), lymphadenopathy, marked eosinophilia, and elevated IgE levels. Since the first reported case in 1937 in China, about 400 cases have been described in the literature, however, some cases of non-Asians with KD have been reported in the last years [121,122,123,124,125,126,127]. The main differential diagnosis is angiolymphoid hyperplasia with eosinophilia (ALHE). Both share significant overlap in clinical symptoms (soft tissue masses in the head and neck region) but can be distinguished by both clinical and histologic features. Systemic manifestations are rarely found in ALHE and therefore kidney involvement is more suggestive for KD [128]. Coexisting renal disease is common in KD and described in about 12–18% of the patients with about two-thirds of them developing nephrotic syndrome [33]. On kidney biopsy, various pathologies including minimal change disease (MCD), mesangial proliferative glomerulonephritis (MsPGN), focal segmental glomerulosclerosis (FSGS), IgM nephropathy, IgA nephropathy, and MN are described [33,129]. MsPGN, MN, and MCD are the most common histological lesions associated with KD [130,131,132]. The etiopathogenesis of KD is still not understood. Considering the regularly present eosinophilia and elevated IL-5 and IgE-levels, an abnormal T-cell stimulation to a hypersensitivity type reaction is suggested [33,133]. Chim et al. showed clonality in T-cell receptor studies in a case of KD. They propose that an initial immunologic response to an unidentified pathogen or stimuli may lead to prolonged lymphoid stimulation and eventually clonal proliferation in some cases, which has been suggested to be a pathogenetic factor of MCD as well [134,135]. First line treatment of KD includes surgical excision and radiation therapy. Oral corticosteroids are used in coexisting renal disease with good response but recurrence of proteinuria after termination of steroid therapy is frequent [33,129,136]. 

Another rare disease combining eosinophilia and kidney injury is tubulointerstitial nephritis and uveitis (TINU) syndrome. It has been described in 1975 by Dobrin et al., while meanwhile about 200 cases have been reported [137,138]. The incidence is highest in children, but all ages may be affected. Patients may initially present with azotemia as first finding and may later develop uveitis, but most cases are diagnosed by an ophthalmologist and abnormal kidney function is found during investigation of the underlying cause, although nephritis typically precedes uveitis in TINU syndrome [139]. Given the high number of cases with ‘idiopathic’ uveitis and the often subtle clinical signs of TIN, it is not surprising that TINU is regarded as a considerably underdiagnosed entity [35]. Mandeville et al. reviewed 133 cases of TINU syndrome and found blood eosinophilia in 17% of patients [35]. This number correlates roughly with the incidence of eosinophilia described in the setting of AIN without uveitis and in both diseases the role of eosinophils in the pathogenesis is unclear. TINU syndrome is thought to arise from a coincidence of certain host susceptibility factors and environmental triggers, leading to an autoimmune-mediated process [138]. Potential HLA risk alleles (depending on geographic background) for TINU, compared to TIN without uveitis or uveitis without TIN, are DRB1*01, DQA1*01, and DQB1*05 [140,141,142,143]. Modified C-reactive protein (CRP) might play a role as a target autoantigen in TINU syndrome [144]. Interestingly, TINU syndrome can recur after kidney transplantation [145]. In general, the degree of AKI in TINU is not severe and renal function often recovers spontaneously. In the large retrospective review by Mandeville et al., 80% of the patients received systemic corticosteroids and 9% further immunosuppressive therapy because of recurrent disease. About half of the patients had recurrence of uveitis, but ophthalmologic prognosis was good. Only 11% of the patients had persistent renal dysfunction, 5 out of 133 patients required dialysis and two of them underwent chronic dialysis [35]. 

IgG4-Related Disease (IgG4-RD) is a recently recognized fibroinflammatory disorder, characterized by IgG4-positive lymphoplasmacytic tissue infiltration, storiform fibrosis, and commonly elevated serum IgG4-levels [146]. Elevated serum IgG4-levels had been associated with autoimmune pancreatitis (AIP) before, but in 2003 a new disease entity was born when systemic manifestations in eight patients with autoimmune pancreatitis were identified and associated with tissue infiltration of IgG4-positive plasma cells [147,148]. IgG4-RD can virtually affect any organ and like AIP, several diseases formerly believed to be organ-limited—like Mikulicz’s syndrome of salivary and lacrimal glands, Ormond’s disease with retroperitoneal fibrosis, or Riedel thyroiditis—now fall within the spectrum of IgG4-RD [146]. The pathogenesis is not fully understood. Although not yet proven, autoimmunity seems to be the initial immunologic stimulus for a Th2 cell mediated immune response [146]. Recently the intracellular protein annexin A11 was identified as an IgG4-recognizing autoantigen. This finding led to the hypothesis that occupational antigens (especially industrial dust, gases, oils, and solvents [149]) or food and animal antigens [150] may cause tissue injuries in target organs with consecutive release of intracellular antigens like annexin A11 that trigger autoimmune response in genetically susceptible individuals (HLA DRB1*04:05-DQB1*04:01) [151,152,153]. The role of the IgG4 antibody in IgG4-RD is complex and some unique characteristics distinguish IgG4 from other antibody subclasses. Because of unstable disulfide bonds between the heavy chains of the IgG4 molecule, an exchange reaction with other antibodies called ‘Fab-arm exchange’ is continuously performed. The resulting asymmetric, bispecific IgG4 molecules are unable to cross-link antigens and to form immune complexes [146,154]. An animal model showed that both IgG1 and IgG4 can induce specific lesions of IgG4-RD, while IgG4 simultaneously appears to have an inhibitory effect on the pathogenic activity of IgG1 [155]. It remains unclear, whether IgG4 itself is the driving force or a compensatory reaction to reduce tissue injury. Regulation of IgG4 production is dependent on Th2 cells and therefore linked to IgE response and eosinophilia. The latter is seen in about 30% of patients with IgG4-RD [37,38,156]. In the setting of allergic reactions, high IgG4 levels reflect an enhanced production of IL-10 and other anti-inflammatory cytokines and therefore are associated with a decrease of allergic symptoms [154]. Two cohort studies examined the prevalence of atopy, eosinophilia and IgE in patients with IgG4-RD. Atopy was present in 31% (22 of 70) in a retrospective US-cohort study and in 43% (22 of 51) patients of a Japanese-cohort which roughly corresponds to the prevalence of allergic-symptoms in the respective population. Interestingly, both studies showed no significant differences in the frequency of eosinophilia, serum levels of IgG4 and IgE or mean concentrations of IgE and IgG4, and peripheral blood cell counts between the allergy groups and the non-allergy groups. These observations support the suggestion that eosinophilia and IgE elevations are inherent characteristics of IgG4-RD immune response and not caused by an underlying allergic condition [37,38]. 

The kidney itself has been proven to be ‘special’ among the organs affected by IgG4-RD. IgG4-related TIN is the leading renal involvement and less frequently IgG4-related MN can be diagnosed. Thus, two different histopathologic types of lesions are common within one organ. Other glomerular lesions associated with IgG4-RD are IgA nephropathy and mild mesangial immune complex glomerulonephritis not otherwise specified [157]. These types of kidney manifestations need to be distinguished from secondary kidney affection in the course of IgG4-RD due to other organ manifestations, like retroperitoneal fibrosis with consecutive postrenal function impairment. The kidney is involved in about 10–15% of the patients with IgG4-RD [36]. Most data about IgG4-related kidney disease (IgG4-RKD) is obtained by a Japanese cohort study of 23 patients and an American (Mayo Clinic) cohort of 35 patients with biopsy-proven IgG4-related TIN [158,159]. Deduced by the collected data, both studies separately proposed diagnostic criteria for IgG4-related TIN, sharing the typical histologic pattern as the only obligatory criterion plus one (Mayo Clinic criteria), respectively two (Japanese Society of Nephrology criteria) other features, including radiologic abnormalities, total IgG and IgG4 serum-levels, and extra-renal organ involvement [159,160]. In both studies, patients were predominantly male with an average age of approximately 65 years. At the time of diagnosis, nearly all patients had extrarenal manifestations. Beside renal dysfunction, typical laboratory findings were elevated total IgG and/or IgG4 levels, hypocomplementemia, PBE, and a positive ANA test [158,159]. Radiographic abnormalities consisted of multiple, bilateral lesions, or masses (69.6% and 78.3%) seen in MRI or CT [158,159]. A recent study showed improved diagnostic sensitivity of these lesions on MRI including diffusion-weighted imaging (DWI) [161]. 

It is important to recognize IgG4-related inflammatory pseudotumor and distinguish it from renal malignant tumor to avoid unnecessary nephrectomy [162]. Another mimicker of IgG4-RKD is AAV, which can be present with elevated serum IgG4 levels and similar organ manifestations compared to IgG4-RD [163]. Moreover, AAV can be associated with both peripheral and tissue eosinophilia, and biopsy specimens may even show IgG4-positive lymphoplasmacytic infiltrations with storiform fibrosis [164]. Constitutional symptoms such as fever, ANCA-positivity and elevated CRP may be helpful to establish the diagnosis of AAV, on the other hand ANCA-positivity does not exclude IgG4-RD [151,165]. ANCA in AAV are most often IgG1 and IgG4 and its speculated that IgG4-RD might trigger AAV in patients with a vulnerable genetic background [165]. RTX might be preferentially used in overlapping syndromes of IgG4-RD and AAV to treat both diseases. On the other side of the clinical spectrum of IgG4-RD, there is a significant overlap with HES. In a comparative case series, Carruthers et al. evaluated cases of IgG4-RD and lymphocyte-variant HES and found elevated serum IgG4-levels (90% and 42%), PBE (26% and 100%), increased serum IgE (60% and 62%) and clonal T-cell receptor gene rearrangement by PCR (52% and 77%) in a significant number of cases in both diseases [166]. 

In HES, kidney involvement is rare, patients often present with cardiac, pulmonary, or cutaneous manifestations and the disease rarely affects more than three organs, whereas IgG4-RD typically shows multiple organ manifestations [166]. A final diagnosis of IgG4-RD is made after exclusion of IgG4-mimicking diseases, principally through pathological evaluation of biopsy specimens [165,167]. Histopathological features on light microscopy are dense lymphoplasmacytic infiltrates of mainly IgG4-positive plasma cells (>10 IgG4-positive plasma cells per high power field, and/or >40% IgG4/IgG-positive plasma cells in the most affected areas), usually accompanied by interstitial fibrosis with a ‘storiform’ pattern. Immunofluorescence microscopy reveals granular tubular basement membrane immune deposits with a positive staining for IgG4, IgG1, IgG3, and C3 complement. In contrast, IgG4-related MN (about 7–10% of cases) does not exhibit the typical fibroinflammatory pattern of IgG4-RD. Instead, granular glomerular capillary wall deposits of IgG (usually with IgG4-dominance) and negative staining for phospholipase A2 receptor (PLA_2_R) are usually seen in IgG4-related MN [168]. Screening for serum antibodies against M-type PLA_2_R, in contrast to most primary MN cases, is negative in its IgG4-related form [169]. Interestingly, primary or idiopathic MN also characteristically shows IgG4 predominance in the glomerular immune complex deposits [168]. Patients with known IgG4-related TIN or IgG4-RD with other organ manifestations who develop proteinuria have a suspected diagnosis of MN, but in absence of already diagnosed IgG4-RD, differential diagnosis of IgG4-related MN can be challenging and effort must be made to rule out primary MN and secondary causes of MN, such as autoimmune disorders (SLE, rheumatoid arthritis, and Sjögren syndrome), infections, neoplasms, or causative drugs [169]. 

The optimal treatment for IgG4-RD is not known due to the lack of RCT evaluating different therapeutic options. In an international symposium among experts, an initial therapy with steroids was proposed for all symptomatic patients. Asymptomatic patients with involvement of important organs like the kidneys, should also receive early steroid therapy to prevent irreversible organ damage. For some subsets of patients, like asymptomatic IgG4-RD with lymphadenopathy or mild submandibular gland enlargement, “watchful waiting” may be appropriate [167]. The typical initial regimen for both IgG4-RD and IgG4-RKD, is an administration of prednisone (e.g., 30–40 mg/day for two to four weeks) with a stepwise tapering depending on the clinical response [167,170,171]. Although the initial response to glucocorticoids is often good, relapse rates after steroid tapering are high and long-term steroid therapy exposes to major adverse side effects [170,172]. Azathioprine, mycophenolate mofetil, and methotrexate are therefore being used frequently as steroid-sparing agents for induction and maintenance therapy but evidence of clinical trials is still lacking [146]. B-cell depleting agents, such as RTX, appear to be an attractive alternative (even without concomitant glucocorticoid therapy) for both induction and maintenance therapy in IgG4-RD and clinical trials are ongoing (ClinicalTrials.gov identifier: NCT01584388) [173,174,175]. For the use of RTX in IgG4-RKD there is even less evidence, but especially in patients with aggressive IgG-RD and renal involvement, treatment with RTX might have beneficial effects in terms of long-term preservation of renal function [171]. Patients with AAV-overlapping disease are possibly a subgroup that is suited for RTX to treat both diseases [163]. The optimal strategy for RTX as maintenance therapy (e.g., need for retreatment or application intervals) is unclear but baseline elevations of IgG4, IgE, and PBE before treatment with RTX predict disease-relapses independently and could be helpful for therapy monitoring [175].

Several other renal pathologies have been associated with PBE in selected cases. In diabetic nephropathy, PBE is rarely present and might be related to a drug-induced hypersensitivity reaction [176,177] or a consequence of an accompanying interstitial nephritis [178,179]. Interestingly, the predictive value of PBE for diagnosing a *Strongyloides* infection in an endemic community appears to be particularly high in patients with type 2 diabetes mellitus [180].

Various forms of renal pathologies may be caused by the human immunodeficiency virus (HIV), either by the infection itself, its treatment or as a complication of the acquired immunodeficiency syndrome (AIDS) [181]. In general, the association of PBE and HIV-infection is complex. PBE is not an infrequent finding among HIV-infected patients but no association with concomitant kidney disease attributable to HIV has been proposed so far. Al Mohajer et al. found PBE in 9.7% (65/671) among antiretroviral therapy (ART)-naive HIV-patients in a single-center cohort [182]. Patients with eosinophilia were more likely to present with skin rash, while treatment to suppress the HIV viral load did not result in the resolution of eosinophilia. A potential parasite-endemic risk should be assessed to rule out *Strongyloides* or *Schistosoma* infection and in the case of suspected infection empirical therapy with ivermectin or praziquantel may be considered [183,184]. Reported cases of HIV-infected patients with PBE and impaired kidney function are usually a consequence of the side effects, toxicity, or hypersensitivity reactions of antiretroviral treatment [185,186,187]. Thomas et al. propose an association of HLA-B*53:01, a prevalent allele among African individuals, with the risk of raltegravir-induced DRESS syndrome [188]. A systematic review of DRESS-cases in HIV-patients found five different antiretrovirals in 35 reported cases [187]. Eosinophilia was seen in 51.4% (18/35) of the cases, while 28.6% (10/35) of the cases presented with kidney affection. Of note, efavirenz (6/35 cases) was associated with the most severe renal failure. These observations need to be confirmed in larger, prospective studies.

For primary glomerulopathies such as MN [131,189,190], IgA nephropathy [191,192,193,194], or FSGS [195,196,197], an association with PBE was mainly found when described in the context of primary eosinophilia-associated diseases like Kimura’s disease, EGPA, IgG4-RKD, or HES.

#### 3.2.5. Chronic Kidney Disease, Dialysis, and Kidney Transplantation

Slightly elevated peripheral blood eosinophil levels without meeting the criteria for PBE were seen in a prospective cohort study comparing leukocyte counts among US veterans with versus without CKD [198]. Among patients with type 2 diabetes, eosinophilia was associated with albuminuria [199]. Solak et al. investigated pruritus among patients with CKD and showed a mean prevalence of eosinophilia in 18.9% of the patients, interestingly without significant differences between CKD Stages 2 to 5. Patients who suffered pruritus had significantly elevated numbers of peripheral blood eosinophils (*p* = 0.005, OR 3.209) [200]. Studies of uremic patients showed bone marrow eosinophilia without elevated PBE, but high circulating levels of ECP. These findings of an abnormal eosinophil homeostasis suggest an accelerated turnover of peripheral eosinophils due to uremia [201,202].

Hemodialysis-associated eosinophilia (HAE) is not uncommon among patients on hemodialysis and is mostly induced by allergy to components of the dialysis circuit [39]. Dialyzer reactions were quite frequent in the 1970s with a reported prevalence of PBE in up to 39% of all patients on maintenance dialysis [40]. Ethylene oxide, the commonly used sterilizing agent of that time, was associated with allergic reactions and HAE [203,204,205]. Traditional cellulosic membranes of cellophane (Kolff dialyzer) or cuprophane (Kiil dialyzer) are derived from cotton and showed limited biocompatibility because of activation of the complement cascade [206]. Using steam and gamma irradiation as modern sterilization techniques instead of ethylene oxide and replacing traditional cellophane and cuprophane membranes by ester modified cellulose or synthetic polymer based membranes, the prevalence of hemodialysis-associated reactions decreased [39]. Modified cellulose membranes primarily provide low-flux dialysis, whereas synthetic membranes are preferred for high-flux dialysis [39]. Even though less frequent, hypersensitivity reactions are also described with modern dialyzer membranes and it was realized that the dialyzer bio-incompatibility depends on the membrane composition [207]. Anaphylactoid reactions within the first minutes of hemodialysis were reported with high-flux polyacrylonitrile AN69^®^ capillary dialyzers [208]. The reported cases were linked to a concomitant ACE inhibitor treatment that might aggravate increased bradykinin concentrations which are reported in patients using polyacrylonitrile membranes [208,209]. Simon et al. showed a higher incidence of anaphylactoid reactions (AR) in patients dialyzed with synthetic membranes than in patients dialyzed with cellulose membranes [210]. More recently, Hildebrand et al. investigated eosinophilia in three hemodialysis facilities among current hemodialysis patients and showed an increase in the prevalence of HAE (defined as greater than 1000 eosinophils/µL) from 1.5% in 2007 (historical cohort of patients dialyzing at the same facilities) to 4.7% in 2012 [39]. ACE inhibitor treatment again correlated with the presence of eosinophilia (41.7% vs. 12.5%, *p* = 0.049). In the follow up period, no difference was seen in the mean number of hospital admission days (12 months follow-up) or in mortality rates (29 months follow-up) between patients on dialysis with or without HAE. Eosinophilia was absent at dialysis initiation in all patients but one, suggesting that eosinophilia can occur in dialysis as a subclinical effect of the dialysis process. The reason for the reported increasing prevalence of HAE remains unclear, but the overall prognosis for asymptomatic eosinophilia among dialysis patients appears to be good [39]. For patients with peri-dialytic symptoms, eosinophilia and also complement C3a and C5a may be indicative parameters for membrane bio-incompatibility. In these cases, changing the hemodialysis membrane might be beneficial [207,211].

PBE in patients undergoing peritoneal dialysis tends to be mild, episodic, and is often associated with peritoneal fluid eosinophilia (PFE) [42]. The latter was mainly reported in the context of eosinophilic peritonitis (EP), a condition defined as an absolute EC in the peritoneal effluent greater than 100/µL, or a relative EC greater than 10% of the total WBC count when the absolute number of eosinophils in the peritoneal effluent is greater than 40/µL [212,213]. In analogue to HAE, the prevalence of eosinophilia associated with peritoneal dialysis decreased within the last decades. Chan et al. studied PFE in a prospective cohort among 23 patients on continuous ambulatory peritoneal dialysis (CAPD) in the 1980s and found PFE in 60.8% of all patients with 57% of those showing PBE [214]. Identified risk factors associated with EP are exposure to vancomycin, infections (fungal or viral) after a recent catheter replacement or treatment with icodextrin, a polymer used to increase ultrafiltration in order to treat volume overload in ESRD [215,216]. More recent data from a prospective study of 48 patients undergoing CAPD suggested a smaller prevalence of PFE of less than 10% and shows no significant correlation between PBE and PFE [41]. PBE is typically detected in asymptomatic patients following catheter replacement and usually resolves spontaneously within a few days [41,213], although the course can be chronic or recurring with permanent changes in membrane function [217]. Considering the favorable prognosis, a steroid treatment in general is not recommended [214]. In severe cases with abdominal symptoms who require further peritoneal dialysis, corticosteroid therapy showed good effects in case reports for both intraperitoneal and systemic application forms [218,219,220]. Montelukast, a leukotriene receptor antagonist, might be a therapeutic option but until now is only reported in one single case [221].

In kidney transplant recipients (KTR), both PBE and tissue eosinophilia are primarily associated with acute allograft rejection (AAR) [43,222,223,224,225,226]. In this context, tissue eosinophilia in the kidney allograft appears to be a specific and sensitive indicator of irreversible and severe acute (vascular) rejection and should be considered as a predictor of poor transplant outcome [223,225,226]. In a retrospective analysis of 71 patients with AAR, Hongwei et al. found a higher mean level of peripheral blood eosinophils in all grades of AAR (1.5–3.0%) than in controls (0.0–0.9%) and PBE was defined as a relative count of ≥4%, which was present in 20–36% of the patients [43]. Weir et al. found a correlation between the percentage of PBE and the severity and outcome of AAR. Patients with peripheral blood eosinophil percentages ≥4% had a higher irreversible rejection rate (37.9% versus 22.4% compared to those with peripheral blood eosinophil percentages <4%, *p* < 0.01). The absolute EC, however, showed no significant correlation [227]. A current single-center renal allograft biopsy study with 1217 KTR confirmed the correlation between the presence of tissue eosinophils and an impaired allograft outcome but detected no PBE in their study cohort [226]. The latter might be explained by a relatively high maintenance dose of 10 mg prednisone per day among all included patients. With respect to the data provided by Hongwei et al. and Weir et al., peripheral blood eosinophil levels appear to be only slightly elevated in AAR and concomitant immunosuppressive therapy with corticosteroids presumably hinders diagnosis.

Two other conditions should be considered as differential diagnosis of eosinophilia in KTR. Firstly, delayed graft function with PBE is suggestive for AIN. Trimethoprim-sulfamethoxazole is one agent frequently prescribed after kidney transplantation that is associated with drug-induced AIN [228,229]. In contrast to AAR, increasing the steroid doses is less effective and the most striking treatment is withdrawal of the offending agent. Drug-induced TIN is a reversible injury that should be considered in the differential diagnosis of renal allograft dysfunction accompanied by eosinophilia. 

A quite uncommon differential diagnosis, but one with a potentially fatal outcome is *Strongyloides* hyperinefction. *Strongyloides stercoralis* is an intestinal nematode prevalent in tropical and subtropical regions, currently believed to infect an estimated 370 million people worldwide [230,231,232]. Due to autoinfection, larvae may persist undetected for years in immunocompetent hosts or cause only slight clinical symptoms. Immunocompromised patients are at risk of a hyperinfection syndrome due to an uncontrolled multiplication of parasites with potentially life-threatening dissemination to all internal organs [231]. A retrospective multicenter study and review of the literature analyzed clinical features of *Strongyloides* hyperinfection in 133 patients and found fever (80.8%), respiratory (88.6%), and gastrointestinal (71.1%) symptoms as the most common clinical manifestations. PBE was observed in only 34.3% of the cases, probably due to underlying immunosuppressive therapy [44]. Immunosuppressive therapy in general and treatment with glucocorticoids in particular, especially when used as high-dose therapy in transplant rejection, is associated with an augmented risk for *Strongyloides* hyperinfection [233]. *Strongyloides* infection in KTR is well documented in endemic regions, but a quantity of reported cases in non-endemic areas is suspected to be donor-derived [45,234,235,236,237,238,239,240]. The American Transplant Society recommends pretransplant *Strongyloides stercoralis* IgG-antibody testing and stool screening for recipients and donors from endemic areas, with gastrointestinal symptoms or with PBE [241]. A survey of physicians found that only 9% of US physicians-in-training could recognize a person in need of screening for strongyloidiasis and treatment errors occurred often among providers unfamiliar with immigrant health (e.g., prescription of empiric corticosteroids) [242]. A retrospective analysis of the Centers for Disease Control and Prevention (CDC) illustrates the deficient screening of high-risk donors and data provided by the New York Organ Donor Network (NYODN) confirms the effectiveness of a targeted donor screening prior to transplantation [238,240]. A recent survey of US Organ Procurement Organizations showed again widespread uncertainty related to donor screening for *Strongyloides stercoralis* and continuing education and advocacy on the importance of targeted donor screening is emphasized by the US Center for Global Health [243]. Nonetheless, eosinophilia, fever, and gastrointestinal and/or respiratory symptoms in a KTR should prompt further diagnostic steps. The current demographic changes due to increased global migratory movements (case reports in Europe are to date restricted to Spain and Italy) demand a certain awareness for screening and therapy approaches [244,245,246]. Sensitivity rates of serology testing are quite high (about 90%), but false-negative results can occur especially in immunocompromised patients. Once hyperinfection is considered, larvae are usually found in stool specimens but invasive methods to obtain tissue specimens by skin biopsy or gastroscopy might be necessary for definite diagnosis [45]. Therapy differs from almost all other causes of eosinophilia hence that intensified immunosuppressive therapy can have fatal consequences. Mortality rates in KTR with *Strongyloides* hyperinfection are up to 50% [247,248]. Antihelmintic therapy with oral ivermectin is the first-line therapy in hyperinfection syndrome and decreasing the steroid dose might be beneficial [45,233]. 

#### 3.2.6. Renal Cell Carcinoma

Renal cell carcinoma (RCC) is in general managed by urologists but nevertheless, suspected masses might be diagnosed by either radiologists or doctors of internal medicine by performing ultrasound or initially detected by the presence of urinary red cells. RCC can induce various paraneoplastic syndromes, such as hypercalcemia, arterial hypertension, or production of erythropoietin [249,250,251]. Paraneoplastic PBE is rarely described in RCC and restricted to case reports of patients with different RCC-subtypes, such as clear cell RCC and chromophobe RCC but mostly appears to reflect an advanced disease phase with widespread metastasis [252,253,254,255,256]. 

Beside RCC, paraneoplastic PBE is described in a variety of other malignancies, such as gastrointestinal tumors, sarcoma, lung cancer, prostate cancer, cervix cancer, and sarcoma [256,257,258,259]. Release of eosinophilopoietic cytokines by the tumor appears to be an underlying mechanism for reactive eosinophil proliferation in these patients [260]. Other authors linked the presence of PBE to necrosis of the tumor or its metastasis [255,256]. In paraneoplastic eosinophilia, tumor-associated tissue eosinophilia (TATE) is distinguished from tumor-associated blood eosinophilia (TABE). Both may occur simultaneously or separately and their association remains unclear [253]. It is noteworthy that tumors with TATE alone are associated with a better prognosis whilst TABE is associated with a poor prognosis and advanced tumor stadium [255].

#### 3.2.7. Eosinophilia on Nephrology Consultation Service

The largest study up to date was published by Diskin et al. in 2011 in form of a prospective cross-sectional study evaluating the prevalence of eosinophilia on a nephrology consultation service [261]. In 2009, 1339 patients on nephrology consultation service of Auburn University (Auburn, AL, USA) had an assessment of the EC. Reasons for consultation were CKD or ESRD in over 50% of cases and no diagnosed case of parasitic infection was seen. The strongest correlation with an increased EC was seen in unspecific conditions like the presence of vascular disease, pleural effusions, and cirrhosis, while a negative correlation was found for total WBC count and beta-blocker use [261]. No correlation between an increased EC and the aforementioned kidney diseases associated with eosinophilia like vasculitis, TIN, or patients on RRT was observed, mainly impacted by the low number of patients in these subgroups. 

## 4. Proposed Approach for the Management of Unexplained Eosinophilia and Acute Kidney Injury

### 4.1. First Step Approach for the Patient with Unexplained Peripheral Blood Eosinophilia

As shown in Figure 3, every patient should undergo a basic assessment. This includes a clinical status with focus on signs of organ involvement and a thorough past medical history assessment that addresses symptoms of organ involvement, exposures, and prior eosinophil counts. The history of exposures should include a travel history, past and current medications including over-the-counter medications, foods, and occupational or recreational exposures. 

Patients with mild or modest PBE (EC < 1500/µL) and no signs of organ involvement should not undergo further evaluation or treatment and can be monitored in intervals of six months. Patients that appear acutely ill with extremely high EC need hospitalization and prompt evaluation. In life-threatening conditions, urgent treatment with high-dose glucocorticoids to lower EC is indicated. To prevent potentially fatal *Strongyloides* hyperinfection syndrome, accompanying empirical prophylaxis with ivermectin 200 µg/kg is recommended in those with potential exposure. Patients with marked HE (EC > 1500/µL) or patients with modest eosinophilia (EC from 500 to 1500/µL) and signs of organ involvement or positive history of exposures should be screened for organ dysfunction. If organ dysfunction is excluded, periodic monitoring should be performed. 

In the case of organ dysfunction, an efficient diagnostic algorithm to exclude serious underlying diseases that require specific treatment is indicated. The WHO suggests the exclusion of secondary (reactive) causes of eosinophilia as the first step [3], including exclusion of allergy/atopy, hypersensitivity conditions, drug reactions, autoimmune diseases, pulmonary eosinophilic diseases, allergic gastroenteritis, and metabolic conditions such as adrenal insufficiency. To exclude an underlying infection, serology for *Strongyloides* species should be performed in all patients. Additional tests, such as repeated ova and parasite testing, stool culture and antibody testing for specific parasites and other infections may be indicated depending on the exposure history. We recommend that additional tests should depend on individual patient characteristics and the results of the initial assessment. Selected patients should be screened for primary or secondary causes of eosinophilia, starting with those investigations that appear more likely to be informative.

### 4.2. Approach for the Patient with PBE and AKI

When PBE and AKI is present, a three-step approach as shown in Figure 4 should be considered. We emphasize that evidence for such recommendations is lacking and to our knowledge, no such proposal has been made to this point. 

As a first step, we recommend a basic assessment to rule out classical eosinophilic disorders, including hematologic neoplasia, parasitic infection, and allergies (see Figure 1). 

After exclusion of classical eosinophilic disorders, a thorough assessment of renal function parameters and screening for PBE-associated kidney diseases is recommended. Distinguishing nephritic and nephrotic syndrome might be helpful to narrow down possible underlying kidney diseases. In the case of unspecific urinary findings, PBE can be a useful diagnostic marker for AIN which is the most frequent entity associated with PBE. We recommend screening for recent changes in the patient’s medication list with a focus set on PPIs, NSAIDs, and antibiotics, but generally any drug can cause AIN. Prompt withdrawal of any potential causative agents is pivotal. In the absence of kidney function recovery within three to five days following discontinuation of potential causative agents, kidney biopsy is recommended and early steroid treatment and more specific immunosuppressive measures should be considered.

## 5. Conclusions

In classical eosinophilic disorders such as HES, in which eosinophils are the driving force of pathogenesis and organ damage, renal involvement is rare and possibly occurs late in the course of disease, e.g., as a consequence of thromboembolism. Nevertheless, a variety of different diseases that typically affect kidney function are associated with PBE. Here, the presence of PBE may be a helpful diagnostic marker for rare or underdiagnosed entities with only subtle clinical signs such as AIN or CES or in overlapping syndromes like EGPA/HES/GPA or IgG4-RD. 

When marked PBE is present, classical eosinophilic disorders such as an underlying hematologic neoplasm, parasitic infections, or allergic conditions need to be excluded first. Patients with AKI and PBE should be investigated for culprit drugs of AIN (frequent). A history of recent vascular interventions should be assessed to evaluate CES (rare). It is noteworthy that both entities may also appear in a slowly progressive form.

In patients undergoing hemodialysis or peritoneal dialysis, eosinophilia might be an indicator for bio-incompatibility of the dialysis material. In KTR, eosinophilia is associated with AAR and portends a poor transplant outcome. Before initiation of high-dose steroid treatment, an infection with *Strongyloides stercoralis* should be excluded to avoid a potentially fatal *Strongyloides* hyperinfection syndrome. The role of eosinophils in immune-mediated diseases like EGPA and IgG4-RD is incompletely understood. Novel insights into EGPA establish the concept of a disease with two different phenotypes. ANCA-positive patients show a vasculitis phenotype with typical renal involvement, whereas the ANCA-negative phenotype shares clinical overlap with HES. Until now, it is unclear if different EGPA-phenotypes should receive tailored therapy, for instance anti-IL-5 therapy with mepolizumab for ANCA-negative EGPA. From our current knowledge, no concrete statement of this treatment approach on renal outcome can be made. Therefore, clinical trials particularly looking at renal disease in the context of EGPA are highly desirable.

## 6. Future Perspectives

Eosinophilia is a frequent finding in patients seen in a general nephrology practice. Taking this into count, there is a clear need to recommend routine differential blood count assessment, at least in the context of a first visit. In our opinion, large, multi-national cohorts should be assessed to describe the frequency of eosinophilia in specific renal diseases. In patients from endemic regions, *Strongyloides* must be ruled out before kidney transplantation and awareness to detect a potential infection following transplantation needs to be increased. With this review, our aim was to summarize eosinophilia in kidney diseases to strengthen further research efforts in this field. Studies are needed to add missing pieces to the puzzles, i.e., if an EC above the normal level before initiation of drugs potentially leading to AIN increases the risk to develop interstitial nephritis. We will particularly focus on diseases leading to nephrotic syndrome and if eosinophilia in this context is predictive of future flares. In diseases with a known association with eosinophilia, further efforts need to be undertaken to define the different phenotypes. As an example, approaches to propose specific biomarkers related to eosinophil-biology have failed to show promising results in EGPA, such as eotaxin-3 or TARC/CCL17 [262]. Phenotyping of these patients is essential to rule out such shortcomings and to draw firm conclusions. 

## Figures and Tables

**Figure 1 jcm-07-00529-f001:**
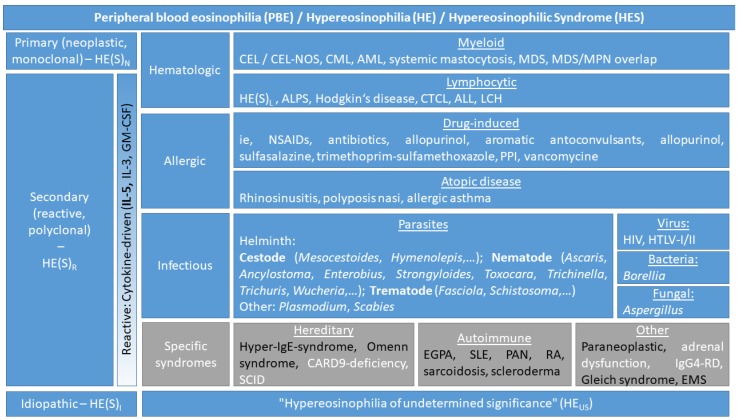
Proposed overview and classification of eosinophilic disorders. Certain specific syndromes (grey boxes) in which the role of eosinophils in the disease mechanism are unclear should be differentiated from ‘real’ HES, according to the International Cooperative Working Group on Eosinophil Disorders (ICOG-EO) proposal (diseases in black letters are explicitly declared as such [2]). Different diseases leading to increased eosinophilic counts are sub-divided into hematologic, allergic, and infectious disorders, while idiopathic hypereosinophilic syndrome is separated from primary and secondary forms. IL, interleukin; GM-CSF, granulocyte-macrophage colony-stimulating factor; HE(S)_M_, myeloid HE(S); HE(S)_L_, lymphocyte-variant HE(S); CEL, chronic eosinophilic leukemia; NOS, not otherwise specified; CML, chronic myeloid leukemia; AML, acute myeloid leukemia; MDS, myelodysplastic syndromes; MPN, myeloproliferative neoplasms; ALPS, autoimmune-lymphoproliferative syndrome; CTCL, cutaneous T-cell lymphoma; ALL, acute lymphocytic leukemia; LCH, Langerhans cell histiocytosis; NSAID, nonsteroidal anti-inflammatory drugs; PPI, proton-pump inhibitor; HIV, human immunodeficiency virus; HTLV, human T-lymphotropic virus; IgE, immunoglobulin E; CARD9, caspase recruitment domain-containing protein 9; SCID, severe combined immunodeficiency; EGPA, eosinophilic granulomatosis with polyangiitis; SLE, systemic lupus erythematosus; PAN, panarteritis nodosa; RA, rheumatoid arthritis; IgG4-RD, IgG4-related disease; EMS, eosinophilia myalgia syndrome.

**Figure 2 jcm-07-00529-f002:**
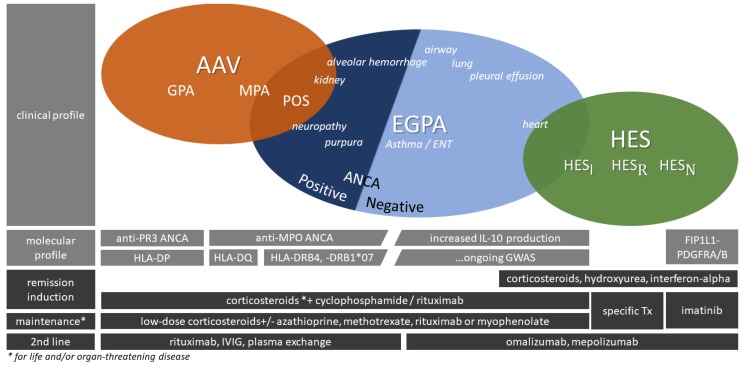
Eosinophil granulomatosis with polyangiitis (EPGA) and overlapping syndromes. The clinical profile of EGPA has distinct but also overlapping symptoms. In the clinical assessment, specific investigations are often needed, since ANCA is negative in most patients. There is a clear overlap between EGPA with a predominant heart involvement and idiopathic hypereosinophilic syndrome (HES_I_). On the other hand, polyangiitis overlap syndrome (POS) has been described for cases that fulfill diagnostic criteria for granulomatosis with polyangiitis (GPA) and EGPA and there is a particular overlap between microscopic polyangiitis (MPA) and EGPA in those with a positive ANCA test presenting with peripheral neuropathy and in some cases kidney and pulmonary disease. A clear distinction is not always possible but should be the aim of the initial investigation. Treatment approaches overlap in some situations. ANCA, anti-neutrophil cytoplasmic antibody; AAV, ANCA-associated vasculitis; GPA, granulomatosis with polyangiitis; MPA, microscopic polyangiitis; POS, polyangiitis overlap syndrome; EGPA, eosinophilic granulomatosis with polyangiitis; ENT, ear, nose, and throat; HES, hypereosinophilic syndrome (I = idiopathic, R = reactive, N = neoplastic); PR3, proteinase 3; MPO, myeloperoxidase; IL-5, interleukin-5; FIP1L1-PDGFRA/B, gene fusion of fibroblast growth factor receptor 1 and platelet-derived growth factor receptor α/β; HLA, human leukocyte antigen; IVIG, intravenous immunoglobulins. According to the EGPA Consensus Task Force, glucocorticoids are the principal therapy to achieve EGPA remission, defined as absence of a clinical systemic manifestation (excluding asthma and/or ear, nose and throat manifestation). In life- and/or organ-threatening disease, remission-induction should include additional immunosuppression (e.g., cyclophosphamide). These patients usually need a maintenance therapy with azathioprine or methotrexate [26]. ANCA-positivity reflects activation of B-cells which could explain the good response seen in studies investigating the effect of RTX [73]. Given the retrospective analyses of RTX efficacy in EGPA, a lower grade of recommendation for the treatment has been issued compared to GPA/MPA according to the EULAR/ERA-EDTA recommendations [74,75,76]. In the largest trial of RTX in EGPA patients (41 cases) until now, Mohammad et al. showed good response rates after one year. Prednisolone could be reduced during follow-up. Interestingly, ANCA-positivity was found to be associated with a higher remission-rate at 12 months compared to ANCA-negative patients (80% vs. 36%) [77]. Other agents employed to reduce corticosteroid doses and maintain remission of EGPA include drugs targeting relevant pro-inflammatory components of the involved immune system. For example, omalizumab (anti-IgE antibody) and mepolizumab (anti-IL-5 antibody) are currently being investigated in several eosinophil disorders like asthma and HES [73]. One randomized controlled trial assigned patients either to mepolizumab or placebo and showed significant higher efficacy rates in the mepolizumab arm, which was achieved in approximately half of the mepolizumab-treated patients [78]—An observation that again adds fuel to the fire in the discussion about different disease phenotypes. Reslizumab and benralizumab are other antibodies targeting IL-5 and its receptor, being tested in phase II trials in patients with eosinophil-driven diseases, including EGPA (ClinicalTrials.gov identifier: NCT02947945; NCT03010436) [79]. In the so-far published clinical trials investigating anti-IL-5 targeted-therapy in EGPA patients, only one patient with glomerulonephritis was enrolled, partly because serum-creatinine elevations >2.5 mg/dL were excluded and renal involvement in EGPA is rare [78,80,81]. Therefore, no firm conclusions from the landmark trial on the effect of mepolizumab on renal impairment can be drawn. In general, renal involvement resembles similar histopathologic features as observed in granulomatosis with polyangiitis (GPA) and microscopic polyangiitis (MPA) with the absence of immune complexes (‘pauci-immune’) in most cases. A study investigating the efficacy of RTX in EGPA recruited patients with either >25% dysmorphic red cells, red cell casts, or biopsy-proven pauci-immune glomerulonephritis. Remission was defined as a composite of stabilization of renal function, absence of active urinary sediment and a significant reduction of the glucocorticoid exposure (<50% of the average dose received over three months before enrollment or <10 mg per day). Following treatment with RTX (375 mg/m^2^/week × 4) patients achieved renal remission which was prolonged during a follow-up period of 12 months [82]. Clinical trials particularly looking at renal disease in the context of EGPA are highly desirable, but recruitment of such patients remains difficult, partly because only one-quarter of EGPA patients present with renal involvement and most are not managed by nephrologists.

**Figure 3 jcm-07-00529-f003:**
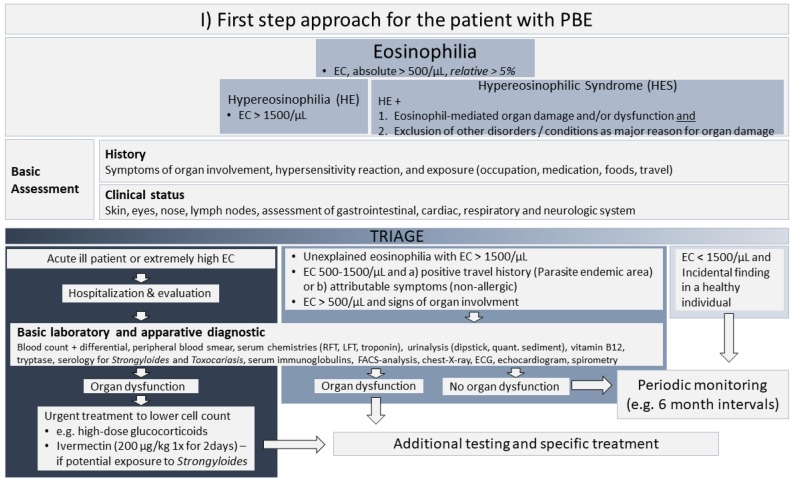
Proposed approach for the initial management of unexplained eosinophilia. When PBE is detected, a systematic management should include a basic assessment, an evaluation of urgency, and the screening for organ dysfunctions. EC, eosinophil count; HE, hypereosinophilia; HES, hypereosinophilic syndrome; RFT, renal function tests; LFT, liver function tests; FACS, fluorescence-activated cell scanning; ECG, electrocardiogram.

**Figure 4 jcm-07-00529-f004:**
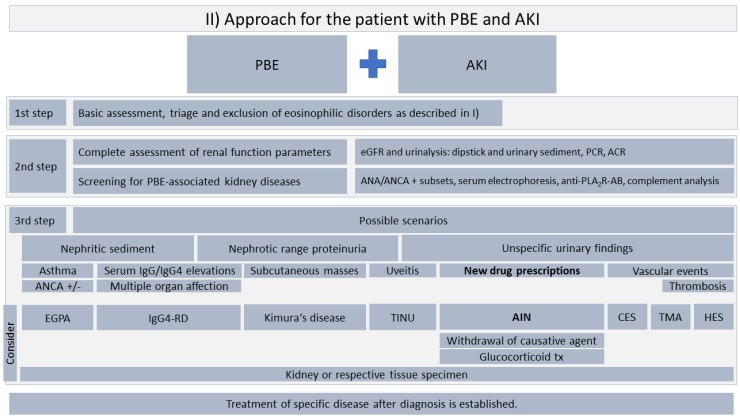
Proposed approach for the management of PBE and AKI. A three-step approach is suggested for the assessment and differential diagnosis. Drug-induced AIN (bold letters) is a comparably frequent finding and prompt identification of the culprit drug is decisive for prognosis. PBE, peripheral blood eosinophilia; AKI, acute kidney injury; HE, hypereosinophilia; EC, eosinophil count; eGFR, estimated glomerular filtration rate; PCR, protein-creatinine ratio; ACR albumin-creatinine ratio; ANA, antinuclear antibody; ANCA, anti-neutrophil cytoplasmic antibody; anti-PLA2R-AB, anti-phospholipase-A2-receptor antibody; IgG, immunoglobulin G; EGPA, eosinophilic granulomatosis with polyangiitis; IgG4-RD, IgG4-related disease; TINU, tubulointerstitial nephritis and uveitis; AIN, acute interstitial nephritis; CES, cholesterol embolization syndrome; TMA, thrombotic microangiopathy; HES, hypereosinophilic syndrome.

**Table 1 jcm-07-00529-t001:** Definition of hypereosinophilia (HE) and the hypereosinophilic syndrome (HES) proposed by the International Cooperative Working Group on Eosinophil Disorders (ICOG-EO).

Term	Definition and Criteria
Peripheral blood eosinophilia (PBE)	>0.5 × 10^9^/L blood
HE	>1.5 × 10^9^/L blood on two examinations (interval ≥4 weeks *) and/or tissue HE defined by the following: Percentage of eosinophils in bone marrow section exceeds 20% of all nucleated cells and/or Pathologist is of the opinion that tissue infiltration by eosinophils is extensive and/or Marked deposition of eosinophil granule protein is found (in the absence or presence of major tissue infiltration by eosinophils).
HES	Criteria for peripheral blood HE fulfilled * Organ damage and/or dysfunction attributable to tissue HE and Exclusion of other disorders or conditions as major reason for organ damage
Eosinophil-associated single-organ diseases	Criteria of HE fulfilled and Single-organ disease

* In the case of evolving life-threatening end-organ damage, the diagnosis can be made immediately to avoid therapeutic delay. Adapted from [2].

**Table 2 jcm-07-00529-t002:** Overview of conditions associated with renal disease and eosinophilia. Cautious interpretation is necessary, since glucocorticoid treatment can mask eosinophilia.

Category	Kidney Disease	% with Renal Involvement	% with PBE	Exclude
a. Hypersensitivity reactions	AIN	100%	App. 22% [19,20]	ATN
	DRESS	11–57% [21,22]	52–95% [23]	
b. Autoimmune disease	EGPA	App. 26%; Higher prevalence (up to 51.4%) with ANCA-positivity [24,25]	100%; Typically, >1500/µL abs. and >10% rel.; Eosinophilic activity may occasionally be organ-limited without blood eosinophilia [26]	MPA, GPA, HES_I_, aspirin-exacerbated respiratory disease, eosinophilic pneumonia, allergic bronchopulmonary aspergillosis, parasitic diseases, allergic conditions, Gleich syndrome, IgG4-RD, diffuse fasciitis with eosinophilia, eosinophilia-myalgia syndrome, eosinophilic myositis, cryoglobulinemic vasculitis, IgA vasculitis
	SLE	app. 50% [27]	<5% [28]	
c. Vascular disease	TMA	App. 77% overall (depending on subtype) [29]	Very rare (number not known)	
	CES	app. 92% [30]	14–71% [31,32]	
d. Other	Kimura’s disease	12–18% [33]	100% [34]	ALHE
	TINU syndrome	100%	app. 17% [35]	Sarcoidosis, Sjögren syndrome, SLE, TB
	IgG4-RD	10–15% [36]	app. 30% [36,37,38]	HES, AAV, malignancies, autoimmune disorders, DI-AIN, MN
e. CKD & RRT	CKD		number not known	
Dialysis	HD-associated eosinophilia		app. 5% (2012) [39], up to 39% in the 1970s [40]	
	PD-associated eosinophilia		<10% (2007) [41], up to 60% in the 1980s [42]	
Kidney transplant	Acute allograft rejection		20–36% (≥4% eosinophils) [43]	(DI-)AIN, *Strongyloides* hyperinfection
	*Strongyloides* superinfection		app. 34% [44]; 67% of pretransplant patients with chronic *Strongyloides* infection [45]	
f. Renal cell carcinoma			Very rare (number not known)	IgG4-RKD

PBE, peripheral blood eosinophilia; AIN, acute interstitial nephritis; ATN, acute tubular necrosis; DRESS, drug reaction with eosinophilia and systemic symptoms; EGPA, eosinophilic granulomatosis with polyangiitis; ANCA, anti-neutrophil cytoplasmic antibody; MPA, microscopic polyangiitis; GPA, granulomatosis with polyangiitis; HES_I_, idiopathic hypereosinophilic syndrome; IgG4-RD, immunoglobulin G4-related disease; CES, cholesterol embolization syndrome; ALHE, angiolymphoid hyperplasia with eosinophilia; TINU, tubulointerstitial nephritis and uveitis; SLE, systemic lupus erythematosus; TMA, thrombotic microangiopathy; TB, tuberculosis; AAV, ANCA-associated vasculitis; DI-AIN, drug-induced-AIN; MN, membranous nephropathy; HD, hemodialysis; PD, peritoneal dialysis; CKD, chronic kidney disease; RRT, renal replacement therapy.

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
