# Peer review of "Eosinophilia and Kidney Disease: More than Just an Incidental Finding?"

_jcm, 2018, doi:10.3390/jcm7120529_

Round 1

Reviewer 1 Report

The current review article provides an overview of different renal pathologies that are associated with Peripheral blood eosinophilia (PBE). The authors should consider following points to improvise the article.

1.       The review article is not comprehensive. Several renal pathologies such as diabetic nephropathy, HIV-associated nephropathy, FSGS, Polycystic kidney disease, and others have not been discussed related to PBE.

2.       The current strategies/treatments/drugs being employed to combat PBE in patients with kidney diseases should be discussed. Also, include a graphical representation of the various therapeutic strategies used to treat PBE.

3.       The authors should include ‘future perspectives’ as a separate heading and highlight the research undertakings to be taken in order to address PBE in kidney patients.

4.       Mention the clinical significance or impact of this review article on research community.

Author Response

Point-to-point response to the reviewer’s comments:

We are very grateful that our manuscript has been reviewed by two external reviewers and our aim during revision was to revise our manuscript accordingly. With these changes, we feel that the manuscript improved.

Minor typographical errors have been eliminated and the manuscript was checked again for linguistic short-comings.

Reviewer 1:

The current review article provides an overview of different renal pathologies that are associated with Peripheral blood eosinophilia (PBE). The authors should consider following points to improvise the article.

1.     The review article is not comprehensive. Several renal pathologies such as diabetic nephropathy, HIV-associated nephropathy, FSGS, Polycystic kidney disease, and others have not been discussed related to PBE.

We are very grateful for this comment. In the section “Other diseases presenting with eosinophilia” a section was added to address this comment. We have added the respective findings in HIV and diabetes mellitus, while association of peripheral blood eosinophilia was mainly related to other pathologies in other nephropathies as stated.

2.     The current strategies/treatments/drugs being employed to combat PBE in patients with kidney diseases should be discussed. Also, include a graphical representation of the various therapeutic strategies used to treat PBE.

For this purpose, we have created section 4 with two additional figures. The aim was to provide an overview regarding general approach when eosinophilia is present. In clinical practice, this might be a helpful path to further classify your patient and initiate treatment in cases when appropriate or to perform further diagnostic steps. Many thanks for this helpful comment. We hope that this particular point improves our comprehensive systematic review.

3.     The authors should include ‘future perspectives’ as a separate heading and highlight the research undertakings to be taken in order to address PBE in kidney patients.

We aim to investigate the association of PBE in patients with nephrotic syndrome and try to associate it with outcome (relapse rate mainly), but eosinophilia has been studied in EGPA for example as disease marker (Dejaco C, et al., as cited in the revised version) as well as eotaxin-3 as potential biomarker, which was initially promising, but has not been corroborated in larger studies). In vitro studies are necessary to study the impact on the respective renal disease. Phenotyping of the patients is a “must” to draw firm conclusions. We have highlighted this in a separate heading as suggested.

4.     Mention the clinical significance or impact of this review article on research community.

First, the relative number of eosinophilia in large nephrology communities needs to be investigated. In order to do so, multi-national investigations are necessary. It is one of our aims to study eosinophilia in much more detail following this review of the literature. We generally (our colleagues and myself) do not comment on eosinophilia during routine clinical follow-up, but there are several patients with “incidental” finding. This issue needs to be addressed and the impact of PBE needs to be defined. We have added this in Section 6 as well and shortly highlighted the need to perform differential blood count at least during the first clinical visit.

Reviewer 2 Report

Dear Authors,

Thank you for paper. I read it, and in my opinion its a very well paper about many connections between peripheral blood eosinophilia and different renal pathologies that are associated with PBE.

Minor comments:

1.     please use the journal style for citations and references;

2.     please try to change the figure 1 - its really hard to understand it;

3.     I like a lot your figure 2, but please try to introduce more explanation about it.

Author Response

Point-to-point response to the reviewer’s comments:

We are very grateful that our manuscript has been reviewed by two external reviewers and our aim during revision was to revise our manuscript accordingly. With these changes, we feel that the manuscript improved.

Minor typographical errors have been eliminated and the manuscript was checked again for linguistic short-comings.

Reviewer 2

Thank you for paper.

I read it, and in my opinion its a very well paper about many connections between peripheral blood eosinophilia and different renal pathologies that are associated with PBE.

We thank Reviewer 2 for this comment. We have observed PBE in several patients during outpatient clinic visits and wondered about this finding. This was the reason to perform this comprehensive review.

Minor comments:

1.     please use the journal style for citations and references;

We have changed the citation/reference style according to your comment.

2.     please try to change the figure 1 – it’s really hard to understand it;

Many thanks for this comment. We changed the Figure 1 according to your comments and hope that it is acceptable in its current form.

3.     I like a lot your figure 2, but please try to introduce more explanation about it;

After your comment, we have added sections to the explanation. It is important to understand this Figure, which is the essential to highlight the complexity of EGPA in our eyes.

Round 2

Reviewer 1 Report

The authors have addressed all the concerns to improvise the review article.